# Beyond SOFA and APACHE II, Novel Risk Stratification Models Using Readily Available Biomarkers in Critical Care

**DOI:** 10.3390/diagnostics15091122

**Published:** 2025-04-28

**Authors:** Jihyuk Chung, Joonghyun Ahn, Jeong-Am Ryu

**Affiliations:** 1Department of Thoracic and Cardiovascular Surgery, Samsung Medical Center, Sungkyunkwan University School of Medicine, Seoul 06351, Republic of Korea; jihyuk.chung@samsung.com; 2Biomedical Statistics Center, Data Science Research Institute, Samsung Medical Center, Seoul 06351, Republic of Korea; jhguy.ahn@samsung.com; 3Department of Critical Care Medicine, Samsung Medical Center, Sungkyunkwan University School of Medicine, Seoul 06351, Republic of Korea; 4Department of Neurosurgery, Samsung Medical Center, Sungkyunkwan University School of Medicine, Seoul 06351, Republic of Korea

**Keywords:** critical care, biomarkers, mortality, prognosis, SOFA score, APACHE II score, lactate-to-albumin ratio, neutrophil percent-to-albumin ratio, risk assessment

## Abstract

**Background:** Current severity scoring systems in intensive care units (ICUs) are complex and time-consuming, limiting their utility for rapid clinical decision-making. This study aimed to develop and validate simplified prediction models using readily available biomarkers for assessing in-hospital mortality risk. **Methods:** We analyzed 19,720 adult ICU patients in this retrospective study. Three prediction models were developed: a basic model using lactate-to-albumin ratio (LAR) and neutrophil percent-to-albumin ratio (NPAR) and two enhanced models incorporating mechanical ventilation and continuous renal replacement therapy. Model performance was evaluated against Sequential Organ Failure Assessment (SOFA) score and Acute Physiology and Chronic Health Evaluation (APACHE) II score using machine learning approaches and validated through comprehensive subgroup analyses. **Results:** Among individual biomarkers, SOFA score showed the highest discriminatory power (area under these curves [AUC] 0.931), followed by LAR (AUC 0.830), CAR (AUC 0.749), and NPAR (AUC 0.748). Our enhanced Model 3 demonstrated exceptional predictive performance (AUC 0.929), statistically comparable to SOFA (*p* = 0.052), and showed a trend toward superiority over APACHE II (AUC 0.900, *p* = 0.079). Model 2 performed comparably to APACHE II (AUC 0.913, *p* = 0.430), while Model 1, using only LAR and NPAR, achieved robust performance (AUC 0.898) despite its simplicity. Subgroup analyses across different ICU types demonstrated consistent performance of all three models, supporting their broad clinical applicability. **Conclusions:** This study introduces novel, simplified prediction models that rival traditional scoring systems in accuracy while offering significantly faster implementation. These findings represent a crucial step toward more efficient and practical risk assessment in critical care, potentially enabling earlier clinical interventions and improved patient outcomes.

## 1. Introduction

The intensive care unit (ICU) is a complex healthcare environment where patients with diverse and critical conditions require intensive treatment and continuous monitoring [1,2]. The dynamic nature of critical illness demands rapid assessment and decision-making to deliver optimal care [3,4]. Conventional severity scoring systems like the Sequential Organ Failure Assessment (SOFA) and Acute Physiology and Chronic Health Evaluation II (APACHE II) are widely used in ICUs to guide clinical decision-making [5,6]. While effective, these systems have limitations, including complexity and the extensive clinical data required for calculation, which can delay assessment and timely intervention [7,8].

To address these challenges, researchers have been investigating simpler methods for predicting patient outcomes. Biomarkers such as the neutrophil-to-lymphocyte ratio (NLR), neutrophil percent-to-albumin ratio (NPAR), lactate-to-albumin ratio (LAR), and urea-to-creatinine ratio (UCR) have shown potential in predicting ICU patient outcomes. These biomarkers offer a more streamlined approach to assessing patient prognosis, often with comparable predictive accuracy to traditional scoring systems [9,10,11,12,13]. However, most studies on these biomarkers have focused on specific conditions rather than the broader spectrum of critically ill patients. This limitation highlights the need for comprehensive research across diverse ICU scenarios.

This study aims to address this gap by investigating the value of various biomarkers in predicting in-hospital mortality across a general ICU population. By comparing these biomarkers with established severity scoring systems, we seek to develop simpler yet broadly applicable indicators. Our goal is to optimize clinical decision-making by providing intensivists with tools that can be easily and quickly applied, potentially improving patient outcomes across various ICU contexts.

## 2. Methods

### 2.1. Study Population

This study was a retrospective, single-center, observational study that included adult patients who were treated at the ICU with SOFA and APACHE II scores evaluated upon ICU admission in Samsung Medical Center, Seoul, Korea between 1 January 2018 and 31 December 2022. The Institutional Review Board (IRB) of Samsung Medical Center approved this study (IRB No. 2024-08-118-001). Informed consent requirements were waived by the Institutional Review Board (IRB) of Samsung Medical Center, given the retrospective nature of this study. This large, single-center cohort was generated in a de-identified form using data extracted by the institutional electronic archive system. The “Clinical Data Warehouse Darwin-C” is an electronic system built for investigators to search and retrieve data from institutional electronic medical records for over 4 million patients with more than 900 million laboratory findings and 200 million prescriptions. Using an extracted raw medical record, independent investigators who were blinded to mortality organized relevant variables of demographic data and underlying diseases. Results of blood laboratory tests and SOFA scores were automatically extracted. Of these patients, those under the age of 18, with incomplete medical records, a “do not resuscitate” order, transfers to other hospitals, or an uncertain prognosis were excluded from this study (Figure 1).

### 2.2. Definitions and Outcomes

LAR was defined as the ratio of lactate concentration (mmol/L) to albumin concentration (g/dL) [10]. Neutrophil-to-lymphocyte ratio (NLR) was defined as the ratio of neutrophil count to lymphocyte count [9,12], and NPAR was defined as the ratio of neutrophil percentage (%) to albumin concentration (g/dL) [13]. UCR was defined as the ratio of the blood levels of urea (BUN) (mmol/L) to creatinine (Cr) (μmol/L) [11]. CRP-to-albumin ratio (CAR) was defined as the ratio of CRP level (mg/L) to albumin concentration (g/dL) [14]. SOFA score was estimated by assessing each component of respiratory, coagulation, liver, cardiovascular, central nervous system, and renal parameters as described in a previous study [6]. The SOFA score was calculated with the worst values recorded during the initial 24 h after ICU admission. APACHE II score was calculated using 12 routine physiological measurements including temperature, mean arterial pressure, heart rate, respiratory rate, oxygenation, arterial pH, serum sodium, serum potassium, serum creatinine, hematocrit, white blood cell count, and Glasgow Coma Scale, along with consideration of age and chronic health conditions [5]. Like SOFA score, APACHE II score was computed using the most abnormal values during the first 24 h of ICU admission. The primary endpoint of this study was the in-hospital mortality.

### 2.3. Model Development and Validation Process

To assess the predictive value of various biomarkers for in-hospital mortality, the study was initiated with a broad set of potential predictors. Utilizing a retrospective cohort of 19,720 adult ICU patients, the data were randomly divided into training (70%) and testing (30%) sets, with this randomization process, repeated 10 times to ensure stability and repeatability of our findings. Initial model development involved constructing logistic regression models within each training set to explore a range of biomarkers and clinical scores, NLR, NPAR, LAR, UCR, CAR, and the established SOFA score. Before finalizing the predictive models, we employed the SHapley Additive exPlanations (SHAP) approach [15], using an XGBoost framework [16], to determine the relative importance of each biomarker and clinical score. This methodology allowed us to quantify the contribution of each variable to the model’s predictive capability. Based on the SHAP values, the most impactful predictors were selected for more focused analysis in subsequent modeling steps. With the significant predictors identified, we refined our logistic regression models to include these variables and then applied these models to the test sets.

### 2.4. Statistical Analyses

Continuous variables are presented as means ± standard deviations, and categorical variables are represented as numbers with subsequent percentages. Data comparison was carried out using Student’s *t*-test for continuous variables, whereas the Chi-square test was performed for categorical variables. The performance of each logistic model was evaluated by constructing receiver operating characteristic (ROC) curves, and the area under these curves (AUC) provided a measure of model discrimination. To statistically compare the performance of these refined models against each other and the traditional SOFA score and APACHE II score, we utilized the DeLong test, a method well suited for comparing the AUCs of two correlated ROC curves. To determine optimal cut-off values for binary classification, Youden’s index was employed. Based on these cut-off values, diagnostic performance metrics including sensitivity, specificity, positive predictive value (PPV), negative predictive value (NPV), and overall accuracy were calculated. All tests were two-sided, and *p*-values of less than 0.05 were considered statistically significant. Statistical analyses were performed with R Statistical Software (version 4.2.0; R Foundation for Statistical Computing, Vienna, Austria).

## 3. Results

From 1 January 2018 to 31 December 2022, a total of 43,109 patients were admitted to the intensive care unit (ICU) of Samsung Medical Center. After excluding patients under 18 years of age and those lacking necessary laboratory data or SOFA scores, 19,720 patients were included in this study. Of these, 666 (3.4%) died during their hospital stay, while 19,054 (96.7%) survived (Figure 1). Comparing the baseline characteristics between the two groups, no statistically significant differences were observed in demographics, comorbidities, or habitual risk factors. However, the non-survivor group showed higher rates of mechanical ventilator use, continuous renal replacement therapy (CRRT), extracorporeal membrane oxygenation (ECMO), and greater administration of vasopressors and inotropic agents (Table 1). This study compared various clinical biomarkers and SOFA scores between survivors and non-survivors. The results revealed significant differences between the two groups in most parameters examined (Table 2).

To validate the robustness of our findings, we performed internal validation through 10-fold cross-validation. The results consistently showed stable performance across all iterations, with minimal variance in AUC values (Figure 2). In the analysis of individual biomarkers (Figure 2A), the SOFA score demonstrated the highest discriminatory power with an AUC of 0.931 (95% CI: 0.922–0.939), followed by the LAR with an AUC of 0.830 (95% CI: 0.814–0.847). Among the biomarker ratios, the CAR and NPAR showed moderate predictive performance with AUCs of 0.749 (95% CI: 0.724–0.774) and 0.748 (95% CI: 0.725–0.771), respectively. The UCR exhibited a lower AUC of 0.652 (95% CI: 0.627–0.676), while the NLR showed the poorest performance with an AUC of 0.497 (95% CI: 0.469–0.525).

Based on the predictive performance analysis and SHAP approach (Appendix A), we identified LAR, NPAR, mechanical ventilation use, and CRRT use as the most significant predictors of in-hospital mortality. Additionally, the logistic regression analysis confirmed that these components were all significantly associated with in-hospital mortality (Table 3). Based on these findings, we constructed three progressive combined models:Model 1: combining LAR and NPAR;Model 2: LAR, NPAR, and mechanical ventilation use;Model 3: LAR, NPAR, mechanical ventilation use, and CRRT use.

The performance analysis of each biomarker and our predictive models demonstrated varying levels of effectiveness in predicting in-hospital mortality (Table 4). Model 3 exhibited high predictive power with an AUC of 0.929 (95% CI: 0.921–0.937), followed by Model 2 (AUC: 0.913, 95% CI: 0.904–0.923), and Model 1 (AUC: 0.898, 95% CI: 0.885–0.911). When compared to established clinical scoring systems, the SOFA score showed an AUC of 0.931 (95% CI: 0.922–0.939), while APACHE II demonstrated an AUC of 0.900 (95% CI: 0.888–0.911). DeLong’s test revealed that Model 3’s performance was statistically comparable to the SOFA score (*p* = 0.052), suggesting similar predictive capabilities for in-hospital mortality. Furthermore, Model 3 showed a trend toward superior performance compared to the APACHE II score (*p* = 0.079). Model 2, while showing lower predictive power than the SOFA score (*p* = 0.003), demonstrated comparable performance to the APACHE II score (*p* = 0.430). Although Model 1 showed relatively lower predictive accuracy compared to both the SOFA score (*p* < 0.001) and APACHE II score (*p* = 0.042), it offers the advantage of simplified implementation. These findings suggest that while Model 3 achieves predictive performance comparable to the established SOFA score and is potentially superior to the APACHE II score, Model 2 presents a practical alternative with prediction capability similar to the APACHE II score. Although Model 1 showed lower predictive power, its simplicity in application could make it a useful tool in resource-limited settings or for rapid initial assessment. Further subgroup analysis across different ICU types demonstrated consistent performance of all three models, suggesting their broad applicability across various critical care settings (Figure 3).

## 4. Discussion

In this comprehensive study of ICU patients, we demonstrated that combined models incorporating simple biomarkers and clinical parameters could achieve predictive performance comparable to or potentially superior to traditional scoring systems for in-hospital mortality. Most notably, our Model 3, which combined LAR, NPAR, mechanical ventilation, and CRRT use, showed comparable performance to the SOFA score and demonstrated a trend toward superior performance compared to the APACHE II score. Even our simpler models showed promising results, with Model 2 achieving performance comparable to the APACHE II score. Notably, the models presented in this study not only exhibited excellent performance and feasibility for predicting clinical outcomes but were also designed with readily accessible and straightforward parameters, making them highly practical for clinical application.

The strong predictive value of LAR and NPAR in our models can be explained by their reflection of key pathophysiological processes in critical illness. LAR serves as an indicator of tissue hypoperfusion and protein synthesis capacity, with elevated lactate levels signaling inadequate tissue oxygenation and decreased albumin reflecting both nutritional status and acute phase response. NPAR, meanwhile, captures both the inflammatory response through neutrophil percentage and the overall metabolic and nutritional state through albumin levels. The synergistic effect of combining these markers provides a comprehensive assessment of both acute stress response and underlying physiological reserve, explaining their robust predictive capability when used together.

The clinical utility of our proposed models lies in their simplicity and accessibility. While SOFA and APACHE II scores require multiple parameters and complex calculations, our models rely on readily available laboratory values and basic clinical information. Model 1, utilizing only LAR and NPAR, can be calculated immediately upon receiving routine admission laboratory results. Even our most comprehensive model (Model 3) requires only the addition of two straightforward clinical parameters—mechanical ventilation and CRRT use. This simplicity allows for rapid risk stratification without the need for complex scoring calculations or specialized measurements, potentially enabling faster clinical decision-making in resource-limited settings.

Our study builds upon and extends previous research in several important ways. While earlier studies have examined individual biomarkers in specific patient populations, our investigation is among the first to evaluate combined models across a large, diverse ICU population. With 19,720 patients from various ICU settings, our study provides robust validation of these biomarkers’ utility. Furthermore, our systematic approach to model development, using SHAP analysis and multiple validation steps, provides strong methodological support for our findings. The consistent performance across different ICU types suggests the broad applicability of our models, addressing a key limitation of previous studies that focused on specific patient subgroups.

Prior studies have typically investigated biomarkers in isolation and within narrower patient populations, limiting their generalizability. For example, Cai C et al. (2022) [13] conducted a study focusing solely on NPAR as a predictor in coronary care unit (CCU) patients, with a cohort of 2364 individuals. Their investigation yielded an AUC of 0.653 (95% CI: 0.623–0.682) for NPAR’s predictive value. In contrast, our models integrate NPAR with other parameters, demonstrating significantly enhanced predictive capability beyond using biomarkers alone.

Similarly, Shin J et al. (2018) [10] evaluated LAR as a predictor of 28-day mortality in sepsis patients, reporting an AUC of 0.69 (95% CI: 0.64–0.73). While their study demonstrated that LAR offered superior predictive capability compared to lactate alone, it was limited to a cohort of 946 sepsis patients. Our investigation extends this work by validating LAR’s utility across a much larger and more diverse ICU population rather than being confined to specific patient subgroups.

Kumrawat A et al. (2024) [17] reported that lactate alone showed better predictive performance for mortality (AUC 0.909) than SOFA (AUC 0.809) or APACHE II scores (AUC 0.769). However, their findings are difficult to generalize due to the limited sample size (200 patients) and results that conflict with other studies in the literature. This inconsistency in the literature highlights a significant advantage of our approach; by combining multiple biomarkers (LAR and NPAR) with indicators of organ support (mechanical ventilation and CRRT), we provide a more comprehensive and generalizable prediction model that overcomes the limitations of single-biomarker studies.

A key innovation in our approach was the strategic incorporation of organ support indicators—specifically mechanical ventilation and continuous renal replacement therapy (CRRT)—alongside biomarkers. These clinical parameters serve as practical proxies for respiratory and/or circulatory dysfunction and renal dysfunction, respectively, which are critical determinants of patient outcomes in ICU settings. By progressively incorporating these organ support indicators into our models, we observed significant improvements in predictive accuracy. This stepwise enhancement demonstrates that while biomarkers alone provide valuable prognostic information, integrating readily identifiable clinical interventions that reflect organ dysfunction substantially improves model performance. Furthermore, these parameters are immediately available at the bedside without additional laboratory testing, making our enhanced models both more accurate and more practically applicable across general ICU populations with diverse underlying conditions.

Several limitations of our study should be acknowledged. First, as a single-center retrospective study, our findings may be influenced by local practice patterns and patient characteristics. The relatively high proportion of postoperative patients (75.7%) in our cohort likely contributed to our relatively low overall in-hospital mortality rate (3.4%), which is lower than typically reported in general ICU populations and may limit generalizability. Additionally, selection bias, inherent in retrospective studies, should be considered when interpreting our results. While our internal validation showed robust results, external validation in different healthcare settings and patient populations is necessary to confirm the broader applicability of our models. Despite these limitations, the consistent performance of our models across different ICU types within our institution suggests promising potential for broader application.

## 5. Conclusions

Our study demonstrates that combined models using readily available biomarkers and clinical parameters can achieve predictive performance comparable to established critical care scoring systems, specifically the SOFA and APACHE II scores. These models offer a simpler, more practical approach to risk stratification while maintaining high predictive accuracy. These findings suggest a promising direction for developing more efficient prognostic tools in critical care, though external validation through multi-center studies encompassing varied healthcare environments and patient characteristics is essential to fully establish their clinical utility. To address this need, we are planning a follow-up project focusing on external validation in a multi-center setting, which will further strengthen the generalizability of our proposed models.

## Figures and Tables

**Figure 1 diagnostics-15-01122-f001:**
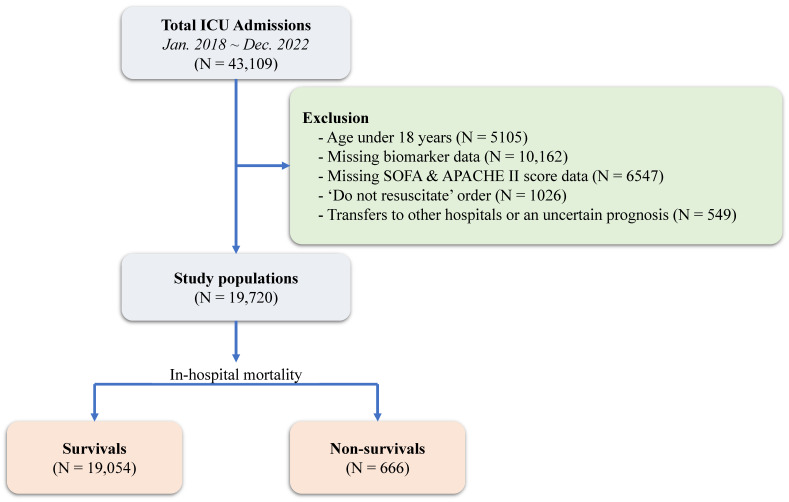
Study flow chart. ICU, intensive care unit; SOFA, Sequential Organ Failure Assessment; APACHE, Acute Physiology and Chronic Health Evaluation.

**Figure 2 diagnostics-15-01122-f002:**
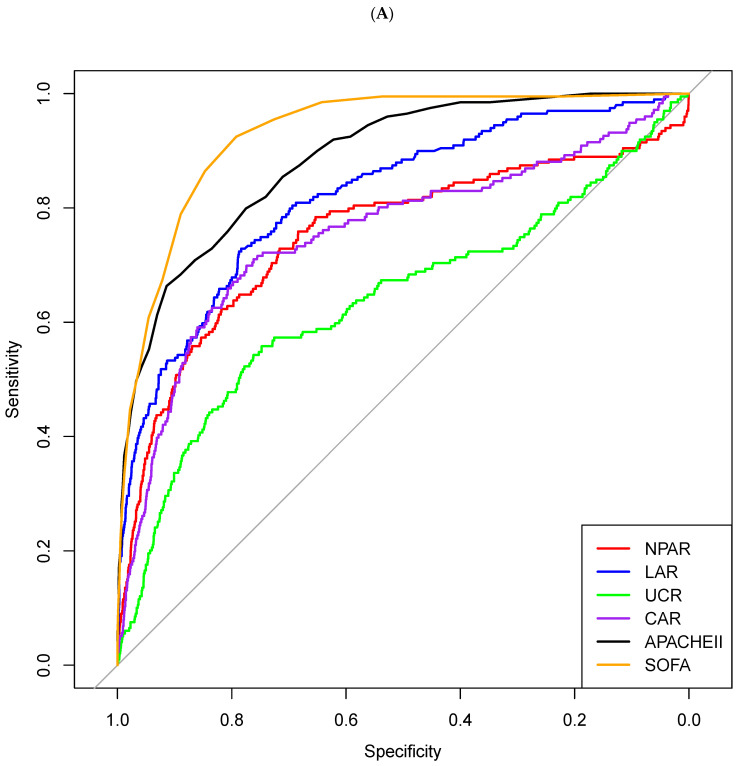
The predictive performance of various biomarkers and combined models for in-hospital mortality was evaluated using receiver operating characteristic (ROC) curves with 10-fold cross-validation. (**A**) Among individual biomarkers, the SOFA score demonstrated the highest discriminative power (AUC = 0.931, 95% CI: 0.922–0.939), followed by LAR (AUC = 0.830, 95% CI: 0.814–0.847), CAR (AUC = 0.749, 95% CI: 0.724–0.774), NPAR (AUC = 0.748, 95% CI: 0.725–0.771), and UCR (AUC = 0.652, 95% CI: 0.627–0.676). (**B**) The ROC curves of three combined prediction models were compared with the SOFA score. Model 1, combining LAR and NPAR, achieved an AUC of 0.898 (95% CI: 0.885–0.911). Model 2, which incorporated LAR, NPAR, and mechanical ventilation, showed improved performance with an AUC of 0.913 (95% CI: 0.904–0.923). Model 3, which added CRRT to the parameters of Model 2, demonstrated the highest predictive capability among the combined models with an AUC of 0.929 (95% CI: 0.921–0.937). All models were validated using 10-fold cross-validation to ensure reliability. SOFA, Sequential Organ Failure Assessment; LAR, lactate-to-albumin ratio; CAR, C-reactive protein-to-albumin ratio; NPAR, neutrophil percent-to-albumin ratio; UCR, urea-to-creatinine ratio; CRRT, continuous renal replacement therapy; AUC, area under the curve; APACHE, Acute Physiology and Chronic Health Evaluation.

**Figure 3 diagnostics-15-01122-f003:**
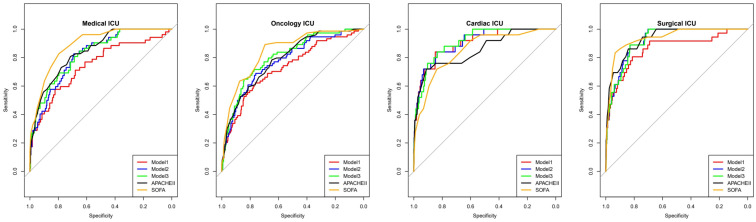
Subgroup analysis of predictive performance across different types of intensive care unit (ICU). SOFA, Sequential Organ Failure Assessment; APACHE, Acute Physiology, and Chronic Health Evaluation.

**Table 1 diagnostics-15-01122-t001:** Baseline characteristics of patients.

	Survivor (*n* = 19,054)	Non-Survivor (*n* = 666)	*p*-Value
Patient demographics			
Age, years	63.0 ± 13.1	62.0 ± 14.2	0.066
Sex, male	12,037 (63.2)	432 (64.9)	0.396
Comorbidities			
Malignancy	11,856 (62.2)	414 (62.2)	1.000
Hypertension	4008 (21.0)	126 (18.9)	0.204
Stroke	2577 (13.5)	86 (12.9)	0.886
Diabetes mellitus	2395 (12.6)	94 (14.1)	0.263
Chronic kidney disease ^a^	1493 (7.8)	49 (7.4)	0.705
Cardiovascular disease	1557 (8.2)	44 (6.6)	0.167
Habitual risk factors			
Alcohol intake	2911 (15.3)	98 (14.7)	0.732
Current smoker	1318 (6.9)	48 (7.2)	0.832
Causes of ICU admission			<0.001
Perioperative management	14,424 (75.7)	65 (9.8)	
Cardiovascular disease	1933 (10.1)	90 (13.5)	
Respiratory distress	1010 (5.3)	257 (38.6)	
Abdominal disorder	381 (2.0)	40 (6.0)	
Neurological disorder	277 (1.5)	26 (3.9)	
Post-cardiac arrest syndrome	144 (0.8)	64 (9.6)	
Others	703 (3.7)	94 (14.1)	
Management in the ICU			
Mechanical ventilation	7089 (37.2)	616 (92.5)	<0.001
Continuous renal replacement therapy	580 (3.0)	229 (34.4)	<0.001
Extracorporeal membrane oxygenation	238 (1.2)	75 (11.3)	<0.001
Use of inotropic agent	1288 (6.8)	134 (20.1)	<0.001
Use of vasopressor	851 (4.5)	75 (11.3)	<0.001

^a^ Chronic kidney disease is defined as either kidney damage or glomerular filtration rate less than 60 mL/min/1.73 m^2^ for 3 months or longer. Data are presented as numbers (%) or means ± standard deviations.

**Table 2 diagnostics-15-01122-t002:** Clinical outcomes according to biomarkers and SOFA score.

	Survivor (*n* = 19,054)	Non-Survivor (*n* = 666)	*p*-Value
ANC (×10^3^/µL)	9.7 ± 5.2	8.9 ± 13.1	0.138
ALC (×10^3^/µL)	1.1 ± 1.3	0.8 ± 1.2	<0.001
Albumin (g/dL)	3.3 ± 0.6	2.7 ± 0.6	<0.001
CRP (mg/dL)	4.7 ± 6.0	12.4 ± 11.1	<0.001
NPAR	19.8 ± 5.3	26.2 ± 9.1	<0.001
LAR	0.8 ± 0.9	3.0 ± 3.1	<0.001
UCR	17.8 ± 8.6	24.3 ± 15.3	<0.001
CAR	1.6 ± 2.3	4.9 ± 4.6	<0.001
APACHE II	19.4 ± 8.2	35.5 ± 8.8	<0.001
SOFA score	3.2 ± 3.2	11.3 ± 4.2	<0.001

Data are presented as numbers (%) or means ± standard deviations. Abbreviations: SOFA, Sequential Organ Failure Assessment; ANC, absolute neutrophil count; ALC, absolute lymphocyte count; CRP, C-reactive protein; NPAR, neutrophil percent-to-albumin ratio; LAR, lactate-to-albumin ratio; UCR, urea-to-creatinine ratio; CAR, C-reactive protein-to-albumin ratio; APACHE, Acute Physiology and Chronic Health Evaluation.

**Table 3 diagnostics-15-01122-t003:** Logistic regression analysis for in-hospital mortality.

Variable	Coefficient	SE	OR (95% CI)	*p*-Value
NPAR	0.0482	0.0066	1.05 (1.04–1.06)	<0.0001
LAR	0.4056	0.0253	1.50 (1.43–1.58)	<0.0001
Use of MV	2.2578	0.1556	9.56 (7.05–12.97)	<0.0001
Use of CRRT	1.5638	0.1067	4.78 (3.87–5.89)	<0.0001

Model performance: C-statistic = 0.929, R^2^ = 0.335, Brier score = 0.027. All variance inflation factor values < 1.2. Abbreviations: SE, Standard Error; OR, odds ratio; CI, confidence interval; NPAR, neutrophil percent-to-albumin ratio; LAR, lactate-to-albumin ratio; MV, mechanical ventilator; CRRT, continuous renal replacement therapy.

**Table 4 diagnostics-15-01122-t004:** Model performances in predicting in-hospital mortality.

	AUC (95% CI)	Cut-Off	Sensitivity	Specificity	PPV	NPV	Accuracy
NPAR	0.754 (0.712–0.796)	22.7	0.678	0.754	0.086	0.986	0.751
LAR	0.823 (0.792–0.854)	1.01	0.716	0.762	0.093	0.988	0.760
UCR	0.651 (0.606–0.696)	21.1	0.510	0.744	0.064	0.978	0.737
CAR	0.759 (0.714–0.805)	1.9	0.704	0.777	0.092	0.988	0.775
APACHE II score	0.903 (0.882–0.924)	30	0.741	0.880	0.174	0.990	0.876
SOFA score	0.933 (0.919–0.947)	6	0.895	0.806	0.137	0.996	0.809
MV use	0.777 (0.757–0.796)	Used	0.925	0.628	0.078	0.996	0.638
CRRT use	0.657 (0.624–0.691)	Used	0.345	0.970	0.278	0.978	0.949
Model 1	0.853 (0.823–0.882)	0.027	0.807	0.760	0.103	0.992	0.762
Model 2	0.894 (0.875–0.912)	0.046	0.827	0.810	0.128	0.993	0.810
Model 3	0.908 (0.891–0.925)	0.039	0.848	0.812	0.133	0.994	0.813

Abbreviations: AUC, area under the curve; PPV, positive predictive value; NPV, negative predictive value; NPAR, neutrophil percent-to-albumin ratio; LAR, lactate-to-albumin ratio; UCR, urea-to-creatinine ratio; CAR, C-reactive protein-to-albumin ratio; APACHE, Acute Physiology and Chronic Health Evaluation; SOFA, Sequential Organ Failure Assessment; MV, mechanical ventilation; CRRT, continuous renal replacement therapy; Model 1, combining LAR and NPAR; Model 2, LAR, NPAR, and mechanical ventilation use; Model 3, LAR, NPAR, mechanical ventilation use, and CRRT use. For Model 1–3, cut-off values were calculated using Youden’s index based on predicted probabilities from each logistic regression model.

## Data Availability

The data used in this study are not publicly available. However, they can be obtained from the corresponding author upon reasonable request.

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
