# Peer review of "Beyond SOFA and APACHE II, Novel Risk Stratification Models Using Readily Available Biomarkers in Critical Care"

_diagnostics, 2025, doi:10.3390/diagnostics15091122_

Round 1

Reviewer 1 Report

Comments and Suggestions for Authors

general evaluation 
The study addresses a significant clinical challenge and conducts it with great quality. The suggested models show excellent performance; the approach is statistically sound. But low mortality rate, selection bias, and single-center design call for careful interpretation. Before general clinical acceptance, external validation in different ICU groups is absolutely crucial. 
advised: Accept with minor corrections, 
1. Broaden the conversation on constraints (generalizability, selection bias, and death rate). 
2. Correct PPV/NPV value differences. 
3. Emphasize in the next projects the importance of outside validation. 
The paper fits the topic of the journal and significantly helps with critical care risk assessment. With minor revision, it will be fit for publication.

Author Response

general evaluation 
The study addresses a significant clinical challenge and conducts it with great quality. The suggested models show excellent performance; the approach is statistically sound. But low mortality rate, selection bias, and single-center design call for careful interpretation. Before general clinical acceptance, external validation in different ICU groups is absolutely crucial. 

Response. We fully agree with your assessment regarding the necessity of external validation across diverse ICU populations. This is indeed crucial for establishing broader clinical applicability of our proposed models. In the current study, we faced limitations that prevented us from conducting external validation, which is why we implemented rigorous internal validation through 10-fold cross-validation to ensure the robustness of our findings. We acknowledge this as an important limitation and have emphasized it accordingly in our discussion section. We are committed to pursuing external validation in future research projects across multiple centers with varied patient populations. We sincerely appreciate your thoughtful review and valuable insights that have helped strengthen our manuscript.

advised: Accept with minor corrections, 
Comments 1. Broaden the conversation on constraints (generalizability, selection bias, and death rate). 

Response 1. Thank you for your advice to expand the discussion on limitations. Following your suggestion, we have revised the discussion section to more comprehensively address the constraints related to generalizability, selection bias, and mortality rate.

Regarding generalizability, we have more clearly explained that our study, being conducted at a single institution, may be influenced by local practice patterns and patient characteristics. In particular, the relatively high proportion of post-operative patients (75.7%) in our cohort reflects our institution's practice pattern, which may differ from other ICU settings and potentially limit the applicability of our findings.

With respect to selection bias, we have elaborated on the potential biases inherent in retrospective studies. Specifically, we have added a discussion about the possible bias introduced by excluding patients who lacked necessary laboratory data or SOFA scores.

Concerning mortality rate, we have emphasized that our overall in-hospital mortality rate (3.4%) is relatively lower than typically reported in general ICU populations. This is likely related to the aforementioned high proportion of post-operative patients. We have added a discussion on how this lower mortality rate might influence the performance evaluation of our prediction models and the potential differences that might arise when applying our models to ICU settings with higher mortality rates.

We believe this broader discussion of limitations will help readers better understand the context and applicability of our research finding

I have added to the limitations section of the manuscript to address the reviewer's comments (page 13, line 81-91)

Several limitations of our study should be acknowledged. First, as a single-center retrospective study, our findings may be influenced by local practice patterns and patient characteristics. The relatively high proportion of post-operative patients (75.7%) in our cohort likely contributed to our relatively low overall in-hospital mortality rate (3.4%), which is lower than typically reported in general ICU populations and may limit generalizability. Additionally, selection bias, inherent in retrospective studies, should be considered when interpreting our results. While our internal validation showed robust results, external validation in different healthcare settings and patient populations is necessary to confirm the broader applicability of our models. Despite these limitations, the consistent performance of our models across different ICU types within our institution suggests promising potential for broader application.

Comments 2. Correct PPV/NPV value differences. 

Response 2. Thank you for pointing out the inconsistencies in the PPV/NPV values in Table 4. We have carefully reviewed all the values and made the necessary corrections to ensure accuracy and consistency throughout the table. To address this issue thoroughly, we recalculated the optimal cut-off values for each model using Youden’s index and re-derived the corresponding PPV and NPV based on these new thresholds. The corrected values now more accurately reflect the predictive performance of each model and biomarker. We appreciate your attention to detail, which has helped improve the overall quality and clarity of our manuscript.

Comments 3. Emphasize in the next projects the importance of outside validation. 
The paper fits the topic of the journal and significantly helps with critical care risk assessment. With minor revision, it will be fit for publication.

Response 3. Thank you for your suggestion to emphasize the importance of external validation in future work. We have revised our conclusion section to more clearly highlight the need for external validation of our prediction models across diverse ICU settings and patient populations. Specifically, we added the following statement: “External validation through multi-center studies encompassing varied healthcare environments and patient characteristics is essential to fully establish the clinical utility of these models. We are planning a follow-up project focusing on external validation in a multi-center setting, which will further strengthen the generalizability of our proposed models.”

We sincerely appreciate your thorough review and constructive recommendations, which have significantly improved the quality of our manuscript. Your expert insights, particularly regarding the limitations of our study and the importance of external validation, have helped us present our findings in a more balanced and contextually appropriate manner. Thank you for your valuable contribution to enhancing this work.

Reviewer 2 Report

Comments and Suggestions for Authors

Dear Authors,

The paper submitted to Diagnostics is an original article. Overall, the scientific objective is relevant and important. In my opinion, this work needs some revision:

  1. “Before finalizing the predictive models, we employed the SHapley Additive exPlanations (SHAP) approach, using an XGBoost framework, to determine the relative importance of each bi-omarker and clinical score”. - You should add a link
  2. Limited Discussion. Compare your results with those of colleagues.
  3. There is no data for the NLR in Table 2 and Figure 2A, but NLR indicated in the description to the table/figure. Correct, please.
  4. Figure 2B – Present the ROC-analysis for mechanical ventilation separately.
  5. Table 4 – What cut-off was used to calculate the AUC for the NPAR, LAR,UCR, CAR, APACHE and SOFA?
  6. The Authors discuss the high predictive power of the results obtained, however, all parameters have a low sensitivity (Table 4). It's worth discussing.
  7. The text of article does not mention the Suppl.Figure. Correct, please. Add the description to the Suppl.Figure.

Author Response

The paper submitted to Diagnostics is an original article. Overall, the scientific objective is relevant and important. In my opinion, this work needs some revision:

Comments  1. “Before finalizing the predictive models, we employed the SHapley Additive exPlanations (SHAP) approach, using an XGBoost framework, to determine the relative importance of each bi-omarker and clinical score”. - You should add a link

Response 1. Thank you for highlighting the need for proper referencing of our analytical methods. We agree that adding citations for the SHAP approach and XGBoost framework would enhance the methodological clarity of our manuscript. We have now added the following references:

For the SHAP approach: Lundberg SM, Lee SI. A Unified Approach to Interpreting Model Predictions. Advances in Neural Information Processing Systems. 2017;30:4765-4774.

For the XGBoost framework: Chen T, Guestrin C. XGBoost: A Scalable Tree Boosting System. Proceedings of the 22nd ACM SIGKDD International Conference on Knowledge Discovery and Data Mining. 2016:785-794.

Additionally, we have included a brief explanation in the methods section about why these approaches were selected for our analysis. This should provide readers with adequate resources to understand our methodological framework.

Comments 2. Limited Discussion. Compare your results with those of colleagues.

Response 2. Thank you for your valuable comment highlighting the limited discussion in our manuscript. We have expanded the Discussion section by conducting a detailed comparison of our findings with previously published studies and clarifying the uniqueness and advantages of our proposed models in the broader academic context.

we have added the following content to enhance the Discussion section.

Cai et al. conducted a study focusing solely on NPAR as a prognostic marker in coronary care unit (CCU) patients, involving a cohort of 2,364 individuals. Their study reported an AUC of 0.653 (95% CI: 0.623–0.682) for predicting mortality using NPAR. In contrast, our study integrates NPAR with other biomarkers and clinical parameters, significantly improving predictive accuracy and highlighting the limitations of using a single biomarker in isolation.

Similarly, Shin et al. evaluated LAR in a cohort of 946 sepsis patients, demonstrating its value in predicting 28-day mortality with an AUC of 0.69 (95% CI: 0.64–0.73). Although their findings confirmed that LAR was superior to lactate alone, their analysis was restricted to a narrowly defined sepsis population. Our study builds upon this by demonstrating the broader applicability of LAR across a large and heterogeneous ICU population, thereby enhancing its generalizability.

Furthermore, Kumrawat et al. (2024) reported that lactate alone showed better predictive performance for mortality (AUC 0.909) than SOFA (AUC 0.809) or APACHE II (AUC 0.769). However, their findings are limited by a small sample size (n=200) and are inconsistent with other reports in the literature. This inconsistency underscores a key strength of our study: rather than relying on a single biomarker, we propose a model that integrates LAR and NPAR with critical clinical indicators such as mechanical ventilation and CRRT. This multifactorial approach enhances predictive power and supports broader clinical applicability.

We have incorporated this comparative analysis into the revised manuscript to better clarify the originality and strengths of our proposed models. Your suggestion has helped us strengthen the academic rigor and contextual relevance of our Discussion section.

Comments 3. There is no data for the NLR in Table 2 and Figure 2A, but NLR indicated in the description to the table/figure. Correct, please.

Response 3. Thank you for highlighting this inconsistency. You are correct that NLR is mentioned in the descriptions of Table 2 and Figure 2A while the actual data is not presented. We apologize for this oversight.

We intentionally replaced NLR with NPAR in our final analysis due to NLR's poor predictive performance and limited clinical utility in our study population. As shown in our results, NLR demonstrated the lowest AUC (0.497) among all evaluated biomarkers, indicating performance no better than chance. However, we failed to remove all references to NLR from the table and figure descriptions during manuscript revisions.

We have now corrected this inconsistency by removing all references to NLR from the descriptions of Table 2 and Figure 2A to ensure alignment with the presented data. Thank you for your careful review that helped us improve the accuracy and consistency of our manuscript.

Comments 4. Figure 2B – Present the ROC-analysis for mechanical ventilation separately.

Response 4. We appreciate the valuable feedback. To evaluate the discriminative ability of individual clinical interventions, we conducted separate ROC curve analyses for mechanical ventilation and continuous renal replacement therapy (CRRT). The AUC for mechanical ventilation alone was 0.776 (95% CI: 0.766–0.787), while CRRT demonstrated an even lower AUC of 0.657 (95% CI: 0.639–0.675). These findings indicate that each intervention, when considered in isolation, has limited predictive power for in-hospital mortality. In contrast, our proposed models—Model 1 (AUC: 0.898), Model 2 (AUC: 0.913), and Model 3 (AUC: 0.929)—which combine biomarker ratios (LAR, NPAR) with these key clinical parameters, achieve significantly higher discriminatory performance. This highlights the advantage of an integrative approach to risk stratification over reliance on any single variable.

Comments 5. Table 4 – What cut-off was used to calculate the AUC for the NPAR, LAR,UCR, CAR, APACHE and SOFA?

Responses to Comments 5 and 6 are addressed together below.

Comments 6. The Authors discuss the high predictive power of the results obtained, however, all parameters have a low sensitivity (Table 4). It's worth discussing.

Response 5, Response 6. Thank you for your insightful comment regarding Table 4.
We believe your concern may have arisen due to the previously reported values of sensitivity and specificity, which were unusually low and high, respectively, and not clinically plausible. These values were the result of an incorrectly defined cut-off value used in the initial analysis. To address this issue, we reanalyzed the data using Youden’s index to determine the optimal cut-off point, and recalculated the diagnostic performance metrics accordingly. The revised cut-off values and corresponding performance indices are now presented in the updated version of Table 4.

The methodology used for this calculation was as follows:

The dataset was randomly split into a 70% training set and a 30% test set. A predictive model was developed using the training set, and its performance was evaluated on the test set. This process was repeated 10 times using different random splits to ensure stability and robustness of the results. The final performance indices were reported as the averages of the results from the 10 iterations. Importantly, for each iteration, the optimal cut-off point was determined using the training set only, and then applied to the corresponding test set to calculate sensitivity, specificity, PPV, NPV, and accuracy.

Once again, we sincerely appreciate your precise feedback. We have revised Table 4 to reflect the corrected results based on this improved methodology.

AUC (95% CI)

Cut-off

Sensitivity

Specificity

PPV

NPV

Accuracy

NPAR

0.754 (0.712 – 0.796)

0.035

0.678

0.754

0.086

0.986

0.751

LAR

0.823 (0.792 – 0.854)

0.026

0.716

0.762

0.093

0.988

0.760

UCR

0.651 (0.606 – 0.696)

0.035

0.510

0.744

0.064

0.978

0.737

CAR

0.759 (0.714 – 0.805)

0.026

0.704

0.777

0.092

0.988

0.775

APACHEII score

0.903 (0.882 – 0.924)

0.060

0.741

0.880

0.174

0.990

0.876

SOFA score

0.933 (0.919 – 0.947)

0.025

0.895

0.806

0.137

0.996

0.809

MV use

0.777 (0.757 – 0.796)

0.042

0.925

0.628

0.078

0.996

0.638

CRRT use

0.657 (0.624 – 0.691)

0.154

0.345

0.970

0.278

0.978

0.949

Model 1

0.853 (0.823 – 0.882)

0.027

0.807

0.760

0.103

0.992

0.762

Model 2

0.894 (0.875 – 0.912)

0.046

0.827

0.810

0.128

0.993

0.810

Model 3

0.908 (0.891 – 0.925)

0.039

0.848

0.812

0.133

0.994

0.813

Comments 7. The text of article does not mention the Suppl.Figure. Correct, please. Add the description to the Suppl.Figure.

Response 7. Thank you for pointing out this oversight. You are correct that the Supplementary Figure showing SHAP values of biomarkers and clinical scores is not referenced in the main text.

We have made the following corrections:

  1. We have added a reference to the Supplementary Figure in the Results section where we discuss the SHAP analysis: "Based on the predictive performance analysis and SHAP approach (Supplementary Figure 1), we identified LAR, NPAR, mechanical ventilation use, and CRRT use as the most significant predictors of in-hospital mortality."
  2. We have enhanced the description of the Supplementary Figure to provide more context about its significance: "Supplementary Figure 1. SHapley Additive exPlanations (SHAP) values for biomarkers and clinical parameters. This figure illustrates the relative contribution of each variable to the prediction model, with higher SHAP values indicating greater importance in predicting in-hospital mortality. The analysis confirms that LAR, NPAR, mechanical ventilation use, and CRRT use were the most influential predictors in our model."

We sincerely thank you for your thorough review and constructive feedback, which have significantly improved the quality and clarity of our manuscript. Your detailed comments have helped us address important methodological aspects, enhance our discussion, and correct inconsistencies. We appreciate the time and expertise you have contributed to strengthen our work, and we believe these revisions have made the paper more valuable to the scientific community.

Round 2

Reviewer 2 Report

Comments and Suggestions for Authors

Dear Authors,

Thank you for your detailed responses to the comments.

The cut-off in Table 4 is very different from the average values of the parameters in Table 2. For example, NPAR 19.8 ± 5.3 and 26.2 ± 9.1 for Survivor and Non-survivor (Table 2), while the cut-off is 0.035 (Table 4). Similarly, APACHEII – 19.4 ± 8.2 and 35.5 ± 8.8, cut-off – 0.060.

Please check if there are any errors or inconsistencies in Table 4.

Author Response

Thank you for your careful observation regarding the cut-off values in Table 4. You are absolutely correct that there was an inconsistency in our previous table. In our previous version, we incorrectly reported the cut-off values as the probability thresholds from the logistic regression models where each parameter was used as an independent variable, rather than the actual parameter values.

We have now corrected Table 4 to show the actual cut-off values of each parameter that correspond to those probability thresholds. For example, the NPAR cut-off is now shown as 22.7, which aligns with the mean values presented in Table 2 (19.8 ± 5.3 for survivors and 26.2 ± 9.1 for non-survivors). Similarly, we have updated the other cut-off values to reflect the actual parameter thresholds.

For Models 1-3, we have retained the probability thresholds (0.027, 0.046, and 0.039, respectively) as these are combined models, and we have noted this in the table footnote for clarity.

We appreciate your attentiveness in identifying this error, which has helped us improve the accuracy and consistency of our manuscript.